# Analysis of clinical outcomes in elderly patients with impaired swallowing function

Keeya Sunata[1,2], Hideki Terai [1]*, Hatsuho Seki[3], Masatsugu Mitsuhashi[4], Yuka Kagoshima[3], Sohei Nakayama[1], Kenichiro Wakabayashi[4], Kaori Muraoka[3], Yukio Suzuki[1,5], Yusuke Suzuki[1]*

1 Department of Respiratory Medicine, Kitasato University Kitasato Institute Hospital, Minato-ku, Tokyo, Japan, 2 Division of Pulmonary Medicine, Department of Medicine, Keio University, School of Medicine, Shinjuku-ku, Tokyo, Japan, 3 Department of Rehabilitation, Kitasato University Kitasato Institute Hospital, Minato-ku, Tokyo, Japan, 4 Department of Otolaryngology, Kitasato University Kitasato Institute Hospital, Minato-ku, Tokyo, Japan, 5 Department of Pharmacology, Kitasato University, Minato-ku, Tokyo, Japan

☯ These authors contributed equally to this work.
* hidekit926@gmail.com (HT); yusuke-s@insti.kitasato-u.ac.jp (YS)

## Abstract

Japan is the world's leading aging society, and increasing medical expenses for elderly people is an urgent issue. Since aspiration pneumonia in elderly people with impaired swallowing function is a huge problem in Japan, their expected long-term clinical course should be clarified. Accordingly, we collected data from 991 elderly ($\geq$75 years old) patients whose swallowing function was evaluated by Kitasato Institute Hospital's speech therapists (January 1, 2010 to December 31, 2017). We analyzed the relationship between swallowing function and the subjects' long-term prognosis. To clarify the prognostic factors of patients with dysphagia, we obtained their clinical information (age, gender, activities of daily living, nutritional status, availability of alternative feeding pathways such as percutaneous endoscopic gastrostomy, and cognitive function). We confirmed 372 death cases and stratified the cases into three groups using Fujishima's swallowing ability grade, which is used to predict elderly people's real-world life expectancy. Results showed the median survival days were 331 and 952 days in Groups I (Grades 1–3, $n = 308$) and II (Grades 4–6, $n = 153$), respectively, whereas the median survival days for Group III (Grades 7–10, $n = 530$) could not be calculated. We conducted a multivariate analysis using the Cox proportional hazards model with Group I, which revealed that initial grade and percutaneous endoscopic gastrostomy were significant prognostic factors for the subjects' long-term survival. Nevertheless, further discussion is necessary, particularly to determine advanced care planning regarding indications for alternative feeding pathways in elderly patients with severe dysphagia, since percutaneous endoscopic gastrostomy could significantly prolong their survival.

## Introduction

According to the Statistics Bureau of the Japanese Ministry of Internal Affairs and Communications, Japan has a rapidly aging society, with increasing life expectancy and decreasing birth

**Data Availability Statement:** All relevant data are within the manuscript.

**Funding:** This study was supported by a research grant from the Health Care Science Institute to H.

T. and a research grant from the Foundation for Total Health Promotion to H.T. The funders did not play any role in the study design, data collection and analysis, decision to publish, or preparation of the manuscript.

**Competing interests:** NO authors have competing interests.

rates [1]. According to a 2017 report, the proportion of the population aged 65 years and over was 27.7% and that of the population aged 75 years and over was 13.8% [2]. In 2016, life expectancy at birth was 80.98 and 87.14 years for men and women, respectively [3]. As a result, medical expenses for the elderly are increasing rapidly [4]. In the 2017 fiscal year, the medical costs to treat later-stage elderly were 16.0 trillion yen, comprising 37.9% of the national medical care expenditure [5]. Since pneumonia is one of the most common causes of death in the elderly, it is a significant cause of the increase in medical expenditures for this population Pneumonia in the elderly has often been correlated with reduced cognitive function and impaired swallowing function [6]. Further, earlier studies have revealed that aspiration pneumonia is common in patients aged 70 years and older; in some studies, 306 among 382 pneumonia patients were diagnosed with aspiration pneumonia [7, 8].

Once pneumonia occurs in the elderly, both the patients and their family members encounter multiple problems, such as ethical issues pertaining to life-prolonging treatments and increase in financial burden [2, 9–11] For effective decision-making, patients and their families require clinical evidence to determine the expected long-term prognosis. However, to date, most studies have only focused on the short-term prognosis or the recurrence rate of readmission for pneumonia; alternatively, they have reported the long-term prognosis in a limited number of patients who have undergone a percutaneous alternative feeding pathway procedure [12]. In other words, these studies do not discuss the prognoses of patients who did not undergo procedures to prolong their lives; however, from an ethical perspective, it is impossible to conduct randomized trials to assess the effectiveness of life-prolonging treatments. Therefore, we collected detailed medical records to obtain real-world data for all the elderly patients who were assessed for the possibility of impaired swallowing function.

The Kitasato Institute Hospital was founded in 1893 and is located in central Tokyo; its location enabled us to collect long-term medical records of a large proportion of patients who were living in the neighborhood. Furthermore, since this hospital cooperates with nearby nursing homes, home doctors, clinics, and university hospitals, we could obtain long-term follow-up data.

The current study reveals the effectiveness of an initial swallowing ability test in assessing elderly patients' expected life span. Furthermore, it reveals a strong correlation between the prognosis and the existence of alternative nutritional pathways in elderly patients with severe dysphagia. Since it focuses on elderly people with impaired swallowing function, the study is expected to help the patients themselves and their families make decisions on care planning by clarifying the expected clinical courses after being diagnosed with severe dysphagia.

## Materials and methods

### Data acquisition

We conducted an observational study of long-term prognoses. The study adhered to the following inclusion (exclusion) criteria: The participants were all aged ≥ 75 years at the time of the study, and their swallowing function had been evaluated by a speech therapist (ST) at Kitasato Institute Hospital from 2010 to 2017. The following clinical data were collected from medical records retrospectively, until December 31, 2018, based on earlier reports [6, 13–15]: age, gender, initial swallowing ability grade (Fujishima's swallowing ability grade), residence before admission, activities of daily living (ADL), existence of an alternative feeding pathway such as percutaneous endoscopic gastrostomy (PEG) or a central venous (CV) port, cognitive function, existence of close relatives as key persons (whether relatives of the third-degree of kinship are involved), and survival outcome. We determined ADL status by retrospectively assessing patients' medical records on their ability to stand on their own without assistance. Regarding

cognitive functioning, the attending physician's judgment (whether the patient had cognitive impairment) was obtained from medical records.

This study was approved by the Ethics Committee of Kitasato University's School of Medicine (approval number #17071). In accordance with the requirements of our institutional review board, participants were not required to provide written consent and were included on an opt-out basis. Information on the study was provided on the research website, and the patients and their relatives could terminate their participation whenever they wanted to. The study was conducted as per the principles expressed in the Declaration of Helsinki.

## Fujishima's swallowing ability grade

At initial evaluation, we classified the cases into three groups according to their Fujishima's swallowing ability grade [16, 17]. Swallowing ability was assessed by the ST once the attending doctor suspected a swallowing dysfunction and consulted the rehabilitation department (Table 1). In this study, we focused on the initial evaluation of swallowing function during the study period, though some of the patients' swallowing function was repeatedly evaluated. In the hospital, swallowing function was evaluated by one of two well-trained STs who shared information with each other. They made their evaluation of Fujishima's swallowing ability grade by combining the results of several tests and observations.

## Statistical analyses

In this study, statistical analyses were performed using Prism version 8 for Mac (GraphPad Software, San Diego) and SPSS 19.0 for Mac (IBM SPSS Statistics, Chicago). Each group's survival time was plotted as a Kaplan-Meier survival curve, and the groups' survival times were compared using the log-rank test. The effects of several variables on survival time were investigated using Cox proportional hazards regression analysis. Serum albumin levels were compared using Student's t-test. The significance level was determined as a $p$-value $< 0.05$.

## Results

### Overall impact of Fujishima's swallowing ability grade on the elderly

Table 2 summarizes the baseline characteristics of patients. We confirmed 372 death cases among the 991 patients who had been evaluated for swallowing function by an ST in our hospital. We classified the cases into three groups according to the subjects' Fujishima's swallowing ability grades at the initial evaluation during the study period: Group I (Grades 1–3, severe dysphagia) with 308 people, Group II (Grades 4–6, moderate dysphagia) with 153 people, and

**Table 1. Fujishima's swallowing ability grade and corresponding number of cases.**

| Grade | Method of Nutrition (*n*) | Groups |
|---|---|---|
| 1 | No therapy indication (*n* = 27) | **I Severe dysphagia** (*n* = 308) |
| 2 | Indirect therapy (*n* = 105) | |
| 3 | Direct and indirect therapy (*n* = 176) | |
| 4 | Minimal intakes for oral satisfaction (*n* = 32) | **II Moderate dysphagia** (*n* = 153) |
| 5 | Meals, once or twice a day (*n* = 36) | |
| 6 | Meals, three times a day with alternative feeding (*n* = 85) | |
| 7 | Dysphagia diet (*n* = 243) | **III Mild dysphagia** + **Normal** (*n* = 530) |
| 8 | Easy chewable diet (*n* = 154) | |
| 9 | Normal food with observation (*n* = 73) | |
| 10 | Normal swallowing ability (*n* = 60) | |

**Table 2. Baseline characteristics of all patients evaluated for their swallowing ability by a speech therapist.**

| Variables | Numbers | |
|---|---|---|
| Age at initial evaluation | 75–103 years (mean 87.18) | |
| Gender (M/F) | 409/582 | |
| Fujishima's grade at initial evaluation (Group I/II/III) | 308/153/530 | |
| Median number of observation days (range) | Group I | 129 (1–2297) |
| | Group II | 110 (1–2345) |
| | Group III | 317 (2–2364) |
| Median (average) number of observation days of patients who survived | Group I | 215 (518) |
| | Group II | 221 (416) |
| | Group III | 382 (595) |
| Outcome (alive/dead) | 619/372 | |

Group III (Grades 7–10, mild dysphagia and normal) with 530 people (Table 2). By using the Kaplan-Meier method, we calculated the median survival days from initial evaluation to death of the cases in each group, which were 331 and 952 days for Groups I and II, respectively. Group III did not reach valid median survival values (Fig 1). Follow-up periods of patients who survived did not differ much among the groups, with an average of 518 days for Group I, 416 days for Group II, and 595 days for Group III. Further, as expected, the prognosis of Group I was worse than the prognoses of Groups II and III.

## Characteristics of elderly patients with severe dysphagia

After obtaining the results of initial analysis, we focused on Group I to elucidate the prognostic factors of elderly patients with severe dysphagia since there were fewer deaths in Groups II and III. Table 3 summarizes the detailed clinical information collected from these patients. Further, we selected the following features as prognostic factors based on earlier reports [6, 13–15]: age, gender, initial swallowing ability grade (Fujishima's swallowing ability grade), residence before admission, ADL, nutritional status (serum albumin), existence of an alternative feeding pathway such as PEG or a CV port, cognitive function, existence of close relatives as key persons, and survival outcome. Since multiple causes of dysphagia such as senility, dementia, and cerebrovascular disease simultaneously exist in some patients, it makes it difficult to collect this information unambiguously. Among the patients with severe dysphagia ($n = 308$), the mean age was 87.2 years, and 264 patients were considered cognitively impaired.

## Initial Fujishima's swallowing ability grade and existence of PEG as significant prognostic factors in elderly patients with severe dysphagia

We performed multivariate analysis using the Cox proportional hazards model to determine the predictors of prognoses among the collected clinical information. Among the results, PEG and the initial Fujishima's swallowing ability grade were identified as significant predictive markers for the prognosis (Table 4, Fig 2A and 2B). There were no other significant prognostic factors in our analysis, which suggests the utility of the swallowing ability function test in predicting the prognosis of elderly patients with severe dysphagia. Interestingly, Fujishima's swallowing ability grade had significant efficiency in predicting the patients' prognosis independently of the existence of an alternative feeding pathway.

The variables included in this analysis are as follows: age at initial evaluation, gender (male or female), initial Fujishima's swallowing ability grade (Grade 3 vs Grade 1 or 2), residence before admission (nursing home or community living), ADL (yes or no), impaired cognitive

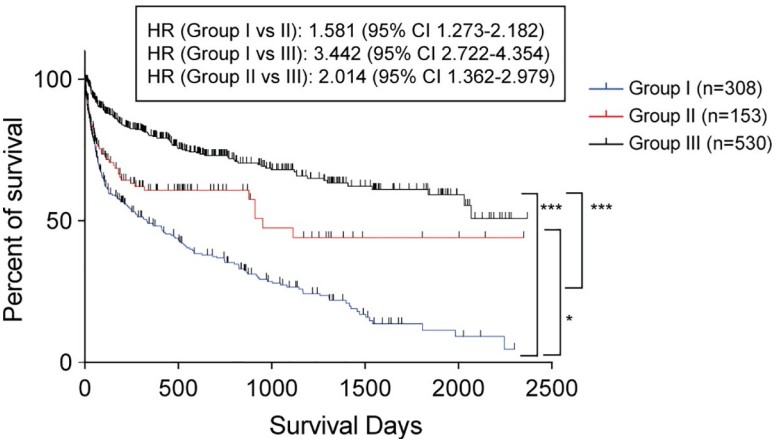

**Fig 1. Kaplan-Meier survival curve of all patients in this study.** Each point indicates days between initial evaluation and death in the death cases. All cases were divided into three groups according to the patients' initial swallowing ability ($^*p < 0.05$, $^{***}p < 0.001$). HR, hazard ratio; CI, confidence interval.

function (yes or no), existence of percutaneous endoscopic gastrostomy (PEG; yes or no), existence of central venous port (yes or no), having key persons of close relatives (yes or no).

## Significant impact of PEG on survival in Grade 1 and 2 patients

We performed a univariate analysis using the Kaplan-Meier method and correlated PEG with the prognosis alone in patients with initial swallowing ability Grades 1 and 2 (Fig 3A and 3B). We could not find any survival benefit of PEG in patients with Grade 3 swallowing ability, which suggests that PEG is necessary for only those patients with very severe dysphagia (Fig 3C). In contrast to the significant life-prolonging effect of PEG in patients with severe dysphagia, a CV port failed to improve the prognosis in our analysis regardless of the existence of PEG (Fig 3D and 3E). Based on the data we could collect, patients with a CV port had poorer nutritional status than did patients without a CV port (serum albumin level was 2.48 [with CV port] vs. 2.88 [without CV port]) (Fig 4). In addition, the initial swallowing grade was strongly

**Table 3. Detailed clinical information of patients in Group I.**

| Variables | Numbers |
|---|---|
| **Age at initial evaluation** | 75–103 years (mean 87.2) |
| | Grade 1 mean 80.8, median 84 |
| | Grade 2 mean 84.9, median 86 |
| | Grade 3 mean 85.7, median 86.5 |
| **Gender (M/F)** | 132/176 |
| **Initial Fujishima's swallowing ability grade (1/2/3)** | 27/105/176 |
| **Residence before admission (nursing home/home)** | 194/114 |
| **Activities of daily living (not able to stand/able to keep standing with assistance)** | 202/106 |
| **Percutaneous endoscopic gastrostomy (yes/no)** | 65/243 |
| **Central venous port (yes/no)** | 19/289 |
| **Impaired cognitive function (yes/no)** | 244/64 |
| **Key persons are close relatives (yes/no)** | 290/18 |
| **Outcome (alive/dead)** | 110/198 |

**Table 4. Results of the Cox regression analysis in Group I patients.**

| Variables | Partial regression coefficient | p-value | Hazard ratio (HR) | 95% CI for HR | |
|---|---|---|---|---|---|
| | | | | Lower | Upper |
| Grade 1 | 0.89 | 0.001 | 2.431 | 1.448 | 4.083 |
| Grade 2 | 0.48 | 0.003 | 1.619 | 1.179 | 2.222 |
| PEG (-) | 0.67 | 0.001 | 1.962 | 1.325 | 2.907 |

CI, confidence interval.

correlated with the prognosis, particularly in patients without any alternative feeding pathway (PEG or a CV port) (Fig 5A and 5B).

## Discussion

The number of deaths from pneumonia has been increasing in Japan because of the increase in proportion and number of elderly people among the nation's population. In most of the elderly patients with pneumonia, swallowing function is impaired and, hence, should be evaluated [2, 8]. Based on our findings, Fujishima's swallowing ability grade helps predict the

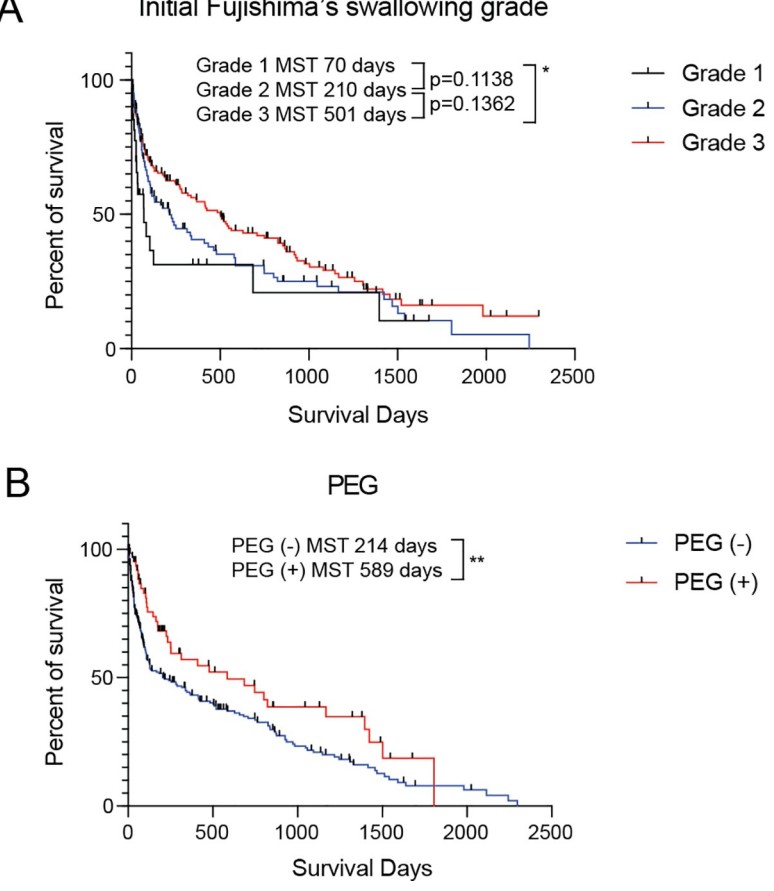

**Fig 2. Group I patients' survival curves classified by significant prognostic factors identified by Cox regression analysis.** (A) Patients were initially classified using Fujishima's swallowing ability grade (*$p < 0.05$). (B) Patients were classified by the existence of PEG (**$p < 0.01$). PEG, percutaneous endoscopic gastrostomy.

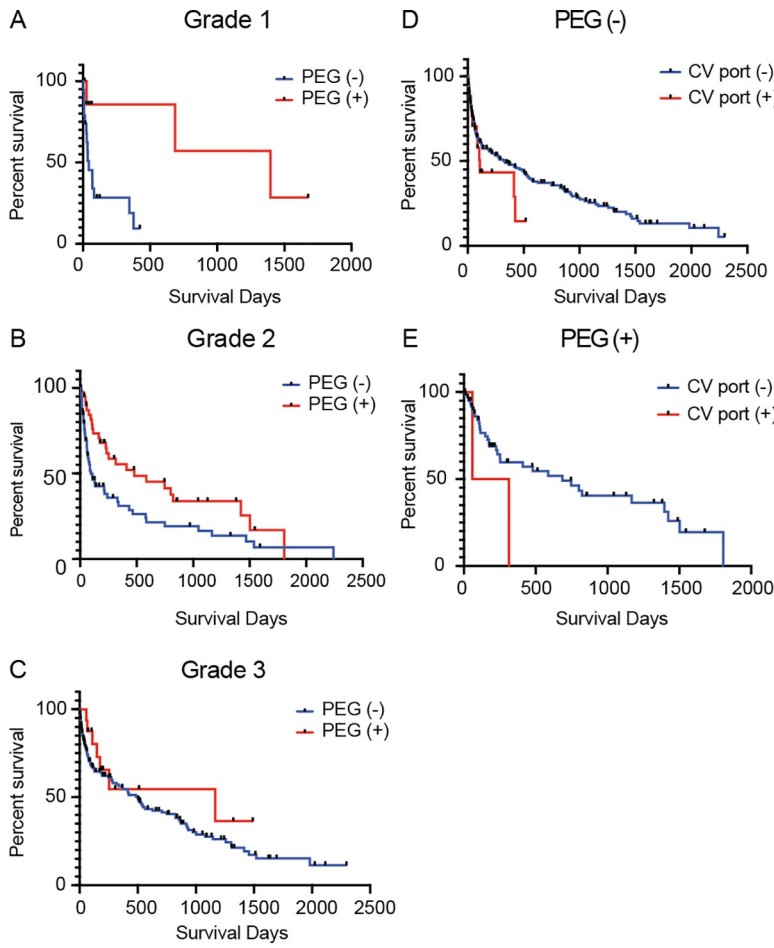

**Fig 3. Assessment of the survival benefit of an alternative feeding pathway (PEG or CV port).** (A), (B), and (C) Survival curves of patients in each initial grade. (D) and (E) Survival curves of patients with or without PEG. PEG, percutaneous endoscopic gastronomy; CV port, central venous port.

prognosis of elderly patients. We have shown that PEG significantly extended life expectancy in patients with very severe dysphagia alone. This observation is consistent with recent reports from Japan [12, 18] but inconsistent with earlier reports from other countries [19, 20]. The discrepancy regarding PEG's utility in elderly patients between the findings of a recent Japanese paper and old reports from other countries can be explained by differences in the patients' background characteristics, health insurance system, or use of sophisticated medical devices [12, 13, 18]. In general, the elderly do not prioritize prolonging their life over ensuring the quality of their life [21–23].

On the other hand, we did not find any survival advantage for the use of a CV port. This is probably because the general condition of patients requiring a CV port is severe. Hence, we should carefully select candidates requiring a CV port with thorough informed consent. Since PEG can improve the prognosis in elderly patients with severe dysphagia, we need to decide the indications for these patients, since euthanasia is not allowed in Japan and, once a life-prolonging device is started, it is very difficult to stop using it. Since most patients with severe dysphagia cannot decide on their medical procedures by themselves because of their decreased cognitive function, their families must face this decision and select appropriate procedures. This issue involves not only a medical problem but also ethical and social problems [24].

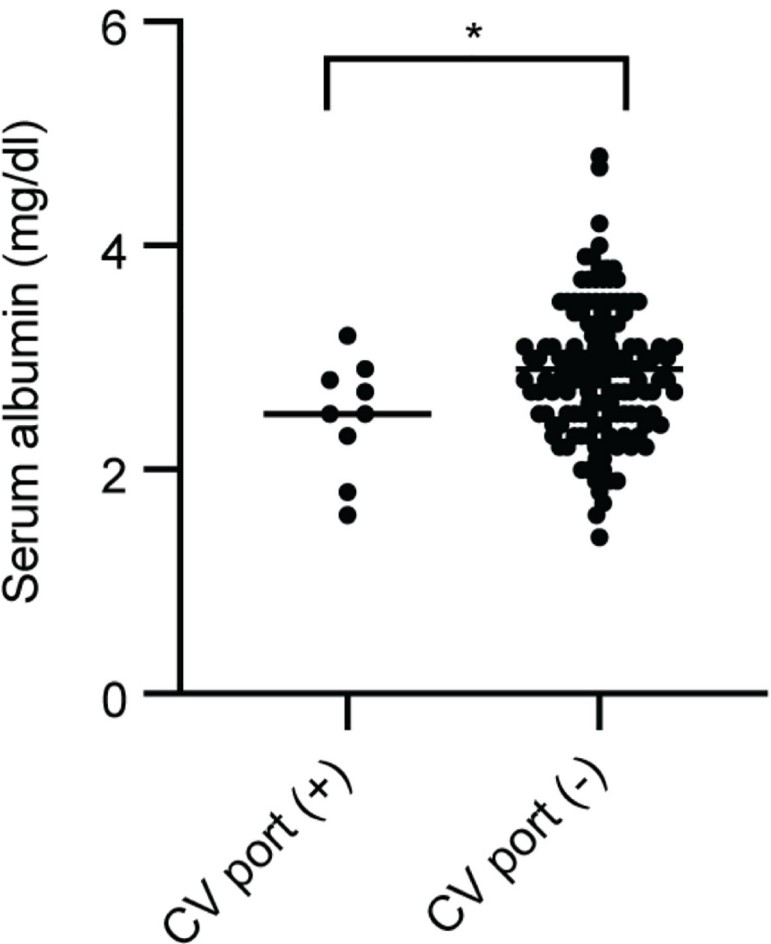

**Fig 4. Correlation of nutritional status (serum albumin concentration) in patients with central venous port.**
Serum albumin level of patients with or without central venous port (CV port). Student's t-test was performed.
$^*p < 0.05$ for patients with CV port (n = 9) versus patients without CV port (n = 142).

It is easy to speculate that lower nutrition levels in patients without oral intake may reduce their survival time. However, no previous study has clearly reported on the limitation of parenteral nutrition, which may increase family's excessive expectation for the survival of the patient. It is often difficult for patients' family or relatives to understand the connection between impaired swallowing function and their survival rate of elderly people. Although previous literature has suggested the utility of percutaneous endoscopic gastrostomy, our study revealed there is a limit, as we could not find any difference in prognosis in Grade 3 patients. Importantly, patients with severe dysphagia will die in a very short time if they are given only peripheral venous nutrition rather than a central venous port. Thus, our data may facilitate better decision-making by family members or close relatives in clinical settings. Furthermore, previous studies have suggested the utility of PEG, but our study revealed there might be a threshold, as we could not find any difference in the prognosis of Grade 3 patients.

The current study has certain limitations, since it was a retrospective study and all the patients' data were obtained from a single institute. Previously, in our institution, one well-trained ST alone had been assessing the swallowing function for all patients since 2008. Since 2010, another ST collaborated with him to evaluate patients. Because the first ST mentored the

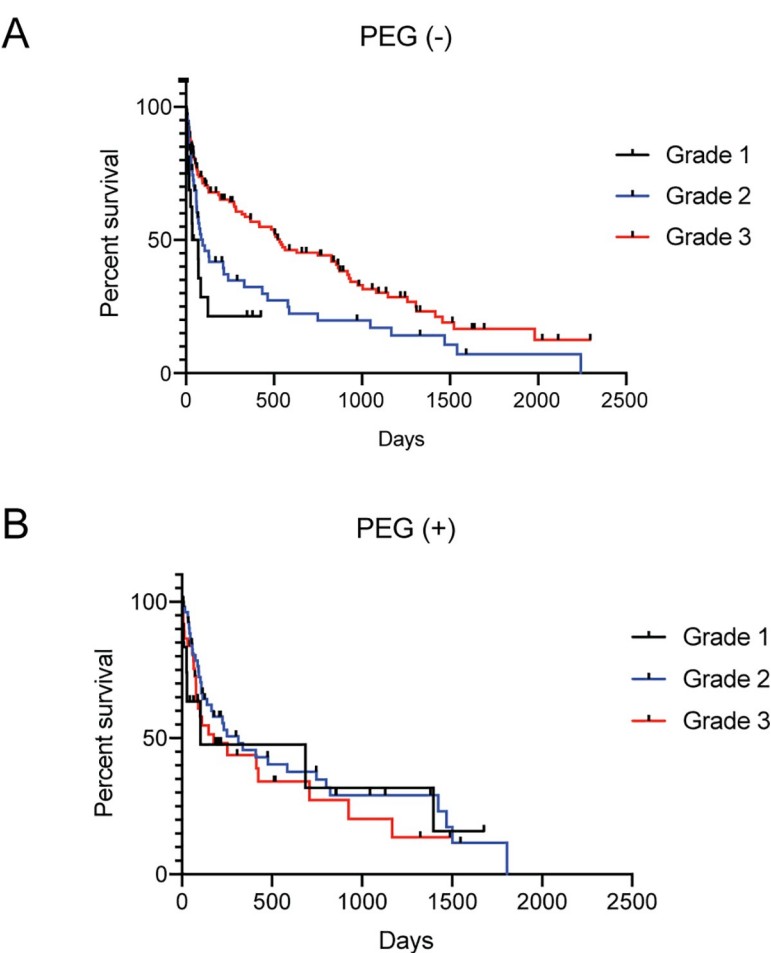

**Fig 5. Initial grade did not predict the prognosis of patients with an alternative feeding pathway.** (A) and (B) Survival curves of patients with or without PEG. Patients were classified into three groups according to their initial grade. PEG, percutaneous endoscopic gastronomy.

second one, their evaluation criteria were similar. Thus, we need to consider the limitation of lack of inter-individual reproducibility.

However, issues surrounding an individual's end of life are very sensitive and cannot be clarified by interventional trials. We successfully collected long-term follow-up data, since this study involved a hospital based in the study area. Despite these limitations, our study is expected to benefit clinical settings and, particularly, individuals involved in the terminal care of elderly people.

## Acknowledgments

We are grateful to Mr. Nomura for his excellent technical assistance. Further, we would like to thank Editage (www.editage.com) for English language editing.

## Author Contributions

**Conceptualization:** Keeya Sunata, Hideki Terai.

**Data curation:** Keeya Sunata, Hideki Terai, Hatsuho Seki, Masatsugu Mitsuhashi, Yuka Kago-shima, Kenichiro Wakabayashi, Kaori Muraoka.

**Formal analysis:** Hideki Terai.

**Visualization:** Hideki Terai.

**Writing – original draft:** Keeya Sunata, Hideki Terai.

**Writing – review & editing:** Keeya Sunata, Hideki Terai, Sohei Nakayama, Yukio Suzuki, Yusuke Suzuki.

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
