## [Decision Letter · Decision Letter 0]

26 May 2020

PONE-D-20-09840

Analysis of clinical outcomes in elderly patients with impaired swallowing function

PLOS ONE

Dear Dr. terai,

Thank you for submitting your manuscript to PLOS ONE. After careful consideration, we feel that it has merit but does not fully meet PLOS ONE’s publication criteria as it currently stands. Therefore, we invite you to submit a revised version of the manuscript that addresses the points raised during the review process.

ACADEMIC EDITOR: In addition to the comments from the reviewers. Please try to add something new, novel, or interesting in the revision to increase the scientific values and the interests of the work.

We look forward to receiving your revised manuscript.

Kind regards,

Jason Chia-Hsun Hsieh, M.D. Ph.D

Academic Editor

PLOS ONE

Additional Editor Comments (if provided):

In addition to the comments from the reviewers. Please try to add something new, novel, or interesting in the revision to increase the scientific values and the interests of the work.

2. Please include a copy of Table 1-2 which you refer to in your text on page 8-9.

Reviewers' comments:

Reviewer's Responses to Questions

**Comments to the Author**

1. Is the manuscript technically sound, and do the data support the conclusions?

Reviewer #1: Partly

Reviewer #2: Partly

2. Has the statistical analysis been performed appropriately and rigorously? 

Reviewer #1: Yes

Reviewer #2: Yes

3. Have the authors made all data underlying the findings in their manuscript fully available?

Reviewer #1: Yes

Reviewer #2: Yes

4. Is the manuscript presented in an intelligible fashion and written in standard English?

Reviewer #1: Yes

Reviewer #2: Yes

5. Review Comments to the Author

Reviewer #1: The study entitled "Analysis of clinical outcomes in elderly patients with impaired swallowing function" is basically well written. However, some methodology information and some results presentations are unclear. Please see my comments below to improve the study.

1. The authors mentioned that they used the data from patients who had been estimated for their swallowing function between 2010 and 2017 to analyze survival hazard. However, it is unclear that when was the end of the present study. That is, when did the authors not follow up the survival status of the patients? If possible, I would also like to know, how many patients were lost to follow-up in the survival status. This is important information because the authors conducted survival analyses.

2. The authors have included several factors in the Cox model. However, the authors only describe how they measured swallow functioning. Some factors may not need to be described (e.g., age and gender). However, some factors need to be introduced because it is hard to know what information was obtained if the authors do not provide such information. For example, what instrument was used for the activities of daily living? Is it Barthel Index? How did the authors assess the cognitive function? There are so many different instruments on cognitive function, especially cognitive function included different aspects (e.g., memory, attention, executive function). Also, who did the assessments on ADL and cognitive function.

3. Although the authors introduced that Fujishima’s swallowing ability grade was used for swallow functions, the authors did not provide detailed information of it. Specifically, what is the psychometric properties of this measure?

4. Following the comment #3, the authors should notify the readers whether the speech therapists have been trained for assessing Fujishima's swallowing ability grade. Also, a limitation should be added if the authors have not information on the inter-rater reliability of the Fujishima's swallowing ability grade in their speech therapists. I assume that there were several speech therapists involved in this study because it is hard to believe that over 7 years, all the patients had their swallowing function assessed by the same therapist. However, I may be wrong and the authors need to explicitly mentioned that there was only one speech therapist in the Methods section if it is real condition.

5. I would recommend moving all the supplementary materials into the main text because to my understanding, Plos One does not have limitations in the number of figures and tables. This will allow potential readers to directly link the text information to figures/tables.

6. The presentation of Table S1 is somewhat mess. Specifically, the title of Grade and its following information are not aligned; the Group III (Mild dysphagia + mild) is not aligned with Groups I and II.

7. From the main text, I understand that Table S2 is presenting a multivariable Cox model. However, the authors only reported significant variables here. Please provide the information of all the included factors in Table S2. Additionally, please explicitly indicate the Grade 3 is the reference group of Grades 1 and 2.

Reviewer #2: This is an interesting and critical study exploring the prognostic factors in severe patients with dysphagia. Authors recruited 991 elderly patients and divided into three groups based on Fujishima’s swallowing ability grade. They investigated the relationship between clinical information and survival days. They described that the prognosis of patients with severe dysphagia was worse than the prognoses of others (moderate or mild dysphagia or normal). Further analysis revealed that Fujishima’s swallowing ability grade and PEG were significant prognostic factors for the long-term survival in severe patients with dysphagia. The concept is very clear and the analysis is appropriate. However, the new findings are very limited and there are several problems in methods, as indicated below.

General comments:

The new findings in the present study are very limited. It is not surprised that the patients with severe dysphagia which represents without oral intake with Fujishima’s grade had longer survival period than the other patients with moderate and mild dysphagia with oral intake. It is easy to imagine lower nutrition level in patients without oral intake than those with oral intake and this is important for survival time, unfortunately authors did not evaluate nutritional status. Previous literatures also suggested that PEG is useful to extend life expectancy. What is the new suggestion in this study compared with previous literatures? I felt the benefits of this study were very few.

This study emphasizes the importance of Fujishima’s swallowing ability grade for the long-term prognosis in patients with dysphagia. I am concerned that the reliability of the assessment of Fujishima’s swallowing ability grade. How to evaluate swallowing function and how to decide the grade? Did authors confirm the reliability of the grade in the present study? Although they described one speech therapist evaluated swallowing function, is this right? I felt this hospital might be not small. If some therapists evaluated swallowing function, the inter-individual reproducibility must be tested.

Authors speculated that the general condition of patients requiring a CV port was severe.

I recommend that they should evaluate subject’s nutritional status.

Why did authors focus on patients with severe dysphagia in the second analysis? How about moderate dysphagia?

Specific comments:

“dysphagic patients” should be replaced with “patients with dysphagia”.

“dysphasia” should be replaced with “dysphagia”.

6. PLOS authors have the option to publish the peer review history of their article (what does this mean?). If published, this will include your full peer review and any attached files.

Reviewer #1: No

Reviewer #2: No

---

## [Author Response · Author response to Decision Letter 0]

30 Jun 2020

PONE-D-20-09840R

Analysis of clinical outcomes in elderly patients with impaired swallowing function

We thank the reviewers for their thoughtful review and interest in our work. Their comments were incredibly insightful and helpful, and we feel that addressing the reviewer’s concerns has significantly strengthened the quality of the manuscript. 

Our point-by-point response is as follows:

Reviewer #1: The study entitled "Analysis of clinical outcomes in elderly patients with impaired swallowing function" is basically well written. However, some methodology information and some results presentations are unclear. Please see my comments below to improve the study.

Thank you very much for your positive comment.

1. The authors mentioned that they used the data from patients who had been estimated for their swallowing function between 2010 and 2017 to analyze survival hazard. However, it is unclear that when was the end of the present study. That is, when did the authors not follow up the survival status of the patients? If possible, I would also like to know, how many patients were lost to follow-up in the survival status. This is important information because the authors conducted survival analyses.

We completely agree with the reviewer. We collected data until December 31, 2018. The median follow-up time of patients who survived was 215 days for Group I, 221 days for Group II, and 382 days for Group III. Similarly, the average follow-up period of patients who survived was 518 days for Group I, 416 days for Group II, and 595 days for Group III. We included these numbers in Table 2. The median follow-up period was shorter for Group I, but we do not think this shorter observation period is the reason for the shorter survival time in Group I since longer observation period usually results in lower survival ratio. 

2. The authors have included several factors in the Cox model. However, the authors only describe how they measured swallow functioning. Some factors may not need to be described (e.g., age and gender). However, some factors need to be introduced because it is hard to know what information was obtained if the authors do not provide such information. For example, what instrument was used for the activities of daily living? Is it Barthel Index? How did the authors assess the cognitive function? There are so many different instruments on cognitive function, especially cognitive function included different aspects (e.g., memory, attention, executive function). Also, who did the assessments on ADL and cognitive function.

We appreciate this important suggestion by the reviewer. Since we retrospectively collected data from medical records, we could hardly obtain precise data on dementia using questionnaires or physical fitness examination for ADLs. Thus, we evaluated ADL according to whether patients could stand without assistance; we determined that a patient had impaired cognitive function by referring to medical records made by the doctor in charge of assessing cognitive function regardless of the method used. Moreover, we collected serum albumin data to assess patients’ nutritional status, although not all the patients were tested for serum albumin.

3. Although the authors introduced that Fujishima’s swallowing ability grade was used for swallow functions, the authors did not provide detailed information of it. Specifically, what is the psychometric properties of this measure?

We used Fujishima’s swallowing ability grade, since it is the most commonly-used scale in clinical settings in Japan. Another grading scale that is widely used in Japan, the Food Intake LEVEL Scale, has almost the same assessment items, and its reliability and validity have been evaluated. The latter is designed for practical use and does not need fluoroscopy or endoscopy for evaluation. We, however, adopted Fujishima’s swallowing ability grade, because it evaluates the ‘ability’ of patients, which seems more appropriate to our study that focuses on the survival period. For our study, two well-trained speech therapists evaluated Fujishima’s swallowing ability using multiple measurements including vocal status, patient posture, oral conditions, results of the modified water swallowing test, and videofluorography or videoendoscopic examination of swallowing. We have added this explanation in the revised manuscript (page 7, lines 125-128).

4. Following the comment #3, the authors should notify the readers whether the speech therapists have been trained for assessing Fujishima's swallowing ability grade. Also, a limitation should be added if the authors have not information on the inter-rater reliability of the Fujishima's swallowing ability grade in their speech therapists. I assume that there were several speech therapists involved in this study because it is hard to believe that over 7 years, all the patients had their swallowing function assessed by the same therapist. However, I may be wrong and the authors need to explicitly mentioned that there was only one speech therapist in the Methods section if it is real condition.

We agree with the reviewer. In our institution, originally, only one well-trained speech therapist began to assess the swallowing function for all patients in 2008. Since 2010, another speech therapist collaborated with him. Because the first speech therapist mentored the second therapist, their evaluation criteria were similar.

5. I would recommend moving all the supplementary materials into the main text because to my understanding, Plos One does not have limitations in the number of figures and tables. This will allow potential readers to directly link the text information to figures/tables.

We agree with the reviewer’s suggestion. We have added all supplementary information to the main text.

6. The presentation of Table S1 is somewhat mess. Specifically, the title of Grade and its following information are not aligned; the Group III (Mild dysphagia + mild) is not aligned with Groups I and II.

Thank you for your comment. We have corrected these errors.

7. From the main text, I understand that Table S2 is presenting a multivariable Cox model. However, the authors only reported significant variables here. Please provide the information of all the included factors in Table S2. Additionally, please explicitly indicate the Grade 3 is the reference group of Grades 1 and 2.

Thank you so much for the comment. We have added all the variables tested to the legend of Table 4. Additionally, we clarified that Grade 3 was used as a reference in the analysis in Table 4.

Reviewer #2: This is an interesting and critical study exploring the prognostic factors in severe patients with dysphagia. Authors recruited 991 elderly patients and divided into three groups based on Fujishima’s swallowing ability grade. They investigated the relationship between clinical information and survival days. They described that the prognosis of patients with severe dysphagia was worse than the prognoses of others (moderate or mild dysphagia or normal). Further analysis revealed that Fujishima’s swallowing ability grade and PEG were significant prognostic factors for the long-term survival in severe patients with dysphagia. The concept is very clear and the analysis is appropriate. However, the new findings are very limited and there are several problems in methods, as indicated below.

Thank you for your positive comments.

General comments:

The new findings in the present study are very limited. It is not surprised that the patients with severe dysphagia which represents without oral intake with Fujishima’s grade had longer survival period than the other patients with moderate and mild dysphagia with oral intake. It is easy to imagine lower nutrition level in patients without oral intake than those with oral intake and this is important for survival time, unfortunately authors did not evaluate nutritional status. Previous literatures also suggested that PEG is useful to extend life expectancy. What is the new suggestion in this study compared with previous literatures? I felt the benefits of this study were very few.

Thank you for your comments. We agree that it is easy to imagine lower nutrition levels in patients without oral intake matters in terms of survival time. We have added this issue to the revised manuscript (page 14, lines 268-271). However, no previous study has presented clinical data to support this idea, and sometimes it is difficult for patients’ relatives to correctly understand the connection between elderly people’s survival rate and impaired swallowing function. Importantly, patients with severe dysphagia would die in a very short time even if they were given a peripheral IV or a central venous port. Thus, our data can help decision-making by relatives in clinical settings. Furthermore, previous literature has suggested the utility of percutaneous endoscopic gastrostomy, but our study revealed there may be a threshold, as we could not find any difference in prognosis in Grade 3 patients. 

This study emphasizes the importance of Fujishima’s swallowing ability grade for the long-term prognosis in patients with dysphagia. I am concerned that the reliability of the assessment of Fujishima’s swallowing ability grade. How to evaluate swallowing function and how to decide the grade? Did authors confirm the reliability of the grade in the present study? Although they described one speech therapist evaluated swallowing function, is this right? I felt this hospital might be not small. If some therapists evaluated swallowing function, the inter-individual reproducibility must be tested.

We agree with the reviewer. We used Fujishima’s swallowing ability grade as it is very commonly used in clinical settings in Japan; however, even though it is common, it has not been published frequently in international journals. For our study, Fujishima’s swallowing function was evaluated by well-trained speech therapists. These therapists usually evaluate Fujishima’s swallowing ability with multiple measurements including vocal status, patient posture, oral conditions, results of the modified water swallowing test, and videofluorography or videoendoscopic examination of swallowing. We have added this explanation to the modified manuscript (page 8, lines 125-128). Further, in our institution, originally, only one well-trained speech therapist began to assess the swallowing function for all patients nine years ago. During the last three years, another speech therapist collaborated with him to evaluate patients. Because the first speech therapist mentored the second one, their evaluation criteria were similar. We discussed this point in the limitations section of this study (page 15, lines 282-288).

Authors speculated that the general condition of patients requiring a CV port was severe.

I recommend that they should evaluate subject’s nutritional status.

We agree with the reviewer’s suggestion. Although there are multiple parameters for nutrition, it is not easy to evaluate nutritional status in a retrospective observational study. We analyzed the relationship between serum albumin and CV port, as it was the only available retrospective data for nutritional status. We found serum albumin levels of 2.48 for patients with a CV port (n = 9) and 2.88 for those without (n = 142), which was added to the result section (page 12, lines 219-221) and Figure 4 in the revised manuscript. 

Why did authors focus on patients with severe dysphagia in the second analysis? How about moderate dysphagia?

We did not find that very many patients with moderate dysphagia had died. So, we considered it difficult to analyze the prognosis of patients with moderate dysphagia. 

Specific comments:

“dysphagic patients” should be replaced with “patients with dysphagia”.

“dysphasia” should be replaced with “dysphagia”.

Thank you so much for your comment. We have corrected these.

Review comments, Reviewer 3:

L. 99~:

This paragraph is quite a few of lacking information in your subjects. I think you should add or rewrite more detail of the subjects' information, such as how Fujishima swallowing ability grade criteria were checked? Who did the exams it, an ST, or some STs? When did you check it? The checking was once and never, or sometimes like every month? And you should put the list of subjects' underlying disease. Generally, the survival days reflects the severity of the underlying disease. And also depending on the cause of the dysphagia, the dysphagia may recover (for example, sequela of cerebral infarction) or may not (for example, neuromuscular disease). It makes no sense to observe only life expectancy without distinguishing the underlying disease. And you only mentioned the age of the subjects "over 75". You should show the age distribution of every group. If subjects were too old at the start point, they could die just even with aging. The details of the cause of death and the breakdown should be shown.

Thank you for your comment. We have collected detailed information of the causes of death and underlying diseases of the patients included in the study. First, we did not have a large number of patients who had died in Groups II and III as compared with Group I. Thus, we only focused on Group I to analyze survival time. Second, elderly people normally have multiple complications, so it is quite difficult to stratify them according to the complications. By reviewing the information from the patients’ medical records, we found that the top three medical conditions for the physician to order evaluation of swallowing function were pneumonia, loss of appetite, and acute-subacute cerebral nerves system disorder. Even patients died because of aging might be determined to have died from such diseases. We also found that the same patient often had multiple reasons simultaneously for the speech therapist consultation. We focused on the initial evaluation of swallowing function because we aimed at providing useful data to informed consent on hospital admission. Specifically, patients’ families sometimes have to decide if percutaneous endoscopic gastrostomy should be performed, and our study can be used as a reference. 

L. 132~:

Although there are supplementary materials, I cannot find Tables 1 and 2.

We apologize for this omission. We have added and modified the manuscript accordingly.

L.185~:

Why your hospital put PEG the patients who were classified grade 3? It seems a bit strange because grade 3 means the patients who has almost enough ability to swallowing.

Thank you for your comment. Patients with Grade 3 cannot start oral intake and just can be trained by food. Even patients with Grade 4 can easy-to swallow food less than the quantity of a meal (enjoyment level) is injested orally. Again, patients with Grade 3 cannot eat enough food; thus, alternative feeding pathway should be considered.

Case Number: 06648983

ref:_00DU0Ifis._5004P1ABcBd:ref

---

## [Decision Letter · Decision Letter 1]

4 Aug 2020

PONE-D-20-09840R1

Analysis of clinical outcomes in elderly patients with impaired swallowing function

PLOS ONE

Dear Dr. terai,

Thank you for submitting your manuscript to PLOS ONE. After careful consideration, we feel that it has merit but does not fully meet PLOS ONE’s publication criteria as it currently stands. Therefore, we invite you to submit a revised version of the manuscript that addresses the points raised during the review process.

ACADEMIC EDITOR: Some minor questions are found in the revised manuscript. 

We look forward to receiving your revised manuscript.

Kind regards,

Jason Chia-Hsun Hsieh, M.D. Ph.D

Academic Editor

PLOS ONE

Additional Editor Comments (if provided):

Some minor questions are found in the revised manuscript. I prefer to recommend a minor revision.

Reviewers' comments:

Reviewer's Responses to Questions

**Comments to the Author**

1. If the authors have adequately addressed your comments raised in a previous round of review and you feel that this manuscript is now acceptable for publication, you may indicate that here to bypass the “Comments to the Author” section, enter your conflict of interest statement in the “Confidential to Editor” section, and submit your "Accept" recommendation.

Reviewer #1: All comments have been addressed

Reviewer #2: All comments have been addressed

Reviewer #3: All comments have been addressed

2. Is the manuscript technically sound, and do the data support the conclusions?

Reviewer #1: Yes

Reviewer #2: Yes

Reviewer #3: Yes

3. Has the statistical analysis been performed appropriately and rigorously? 

Reviewer #1: Yes

Reviewer #2: Yes

Reviewer #3: Yes

4. Have the authors made all data underlying the findings in their manuscript fully available?

Reviewer #1: Yes

Reviewer #2: Yes

Reviewer #3: No

5. Is the manuscript presented in an intelligible fashion and written in standard English?

Reviewer #1: Yes

Reviewer #2: Yes

Reviewer #3: Yes

6. Review Comments to the Author

Reviewer #1: The authors have satisfactorily responded to the previous comments made by both reviewers. I am comfortable to recommend publication for the present form.

Reviewer #2: The authors responded to all reviewer comments.

I am happy to recommend this paper for publication.

Reviewer #3: 

Thank you for correcting the manuscript. The manuscript is very readable and useful. However, the reviewer has still some concerns.

L50 The speech therapist (ST) is common in Japan, but the speech language pathologist (SLP) is common in other countries. Please confirm the title.

L104 Were there any participants who constructed PEG or CV during the observation?

L122 The reviewer is still concerning what was the cause of dysphagia in this population. The prognosis of the disease (whether it is a progressive disease or not) is an important indicator for PEG construction. Information about the approximate cause of dysphagia of this population should be provided to the reader.

L145 The number of deaths is 373 in the text, but it is 372 in Table 1. The number of Grade I people is 308 in the text and Table 1, but in Figure 1 it is 309.

Table 4 "impaired cognitive function" is written twice in the legend.

7. PLOS authors have the option to publish the peer review history of their article (what does this mean?). If published, this will include your full peer review and any attached files.

Reviewer #1: No

Reviewer #2: No

Reviewer #3: No

---

## [Author Response · Author response to Decision Letter 1]

19 Aug 2020

PONE-D-20-09840R

Analysis of clinical outcomes in elderly patients with impaired swallowing function

We thank Reviewers #1 and #2 for recommending our paper for publication. We also appreciate Reviewer #3’s thoughtful feedback, which helped us enrich the quality of our manuscript.

Our point-by-point response is as follows:

Reviewer #1: The authors have satisfactorily responded to the previous comments made by both reviewers. I am comfortable to recommend publication for the present form.

We thank the reviewer for their positive comment and careful review, which helped improve the manuscript.

Reviewer #2: The authors responded to all reviewer comments. I am happy to recommend this paper for publication.

We thank the reviewer for their positive comment and careful review, which helped improve the manuscript.

Reviewer #3: 

1. L50 The speech therapist (ST) is common in Japan, but the speech language pathologist (SLP) is common in other countries. Please confirm the title.

All speech therapists in Japan have passed the National License Examination for Speech-Language-Hearing Therapists. It is the only national certificate which officially prove the skill to engage in speech therapy in Japan and translated in English as ‘speech therapist’. We described the title of the therapists as ‘speech therapist’ because they possess the Japanese license which is not officially translated as ‘speech language pathologist’.

2. L104 Were there any participants who constructed PEG or CV during the observation?

Survival time of patients with PEG might be elongated by patients whose PEG was made during the observation, which suggests better health condition that did not need immediate construction of a parenteral route. Most of the patients with PEG or CV were made the parenteral nutritional route soon after their initial evaluation. Only five patients with severe dysphagia were constructed PEG 100 days or more after the initial evaluation. We analyzed survival time again after excluding those patients. The results of the analysis were the same (log-rank test: p < 0.05). Thus, we wish to retain the original analyses unchanged because the definition of ‘during the observation’ is not always clear.

3. L122 The reviewer is still concerning what was the cause of dysphagia in this population. The prognosis of the disease (whether it is a progressive disease or not) is an important indicator for PEG construction. Information about the approximate cause of dysphagia of this population should be provided to the reader.

We previously tried to divide patients by the causal disease of dysphagia such as brain infarction and by looking up the relevant ICD-10 codes. There were 801 pneumonia-related codes and 619 brain-related codes in the 912 patients. Then, we realized that a single patient often had multiple diseases, which made it difficult to presume the cause of dysphagia. Simple aging might not even be on the list of disease codes. Consequently, we focused on the direct relationship between swallowing function and survival time. For additional clarity, we added the following to the revised manuscript (page 10, lines 183-7 in the revised manuscript): “Since multiple causes of dysphagia such as senility, dementia, and cerebrovascular disease simultaneously exist in some patients, it makes it difficult to collect this information unambiguously. Among the patients with severe dysphagia (n = 308), the mean age was 87.2 years, and 264 patients were considered cognitively impaired.”

4. L145 The number of deaths is 373 in the text, but it is 372 in Table 1. The number of Grade I people is 308 in the text and Table 1, but in Figure 1 it is 309.

Thank you for your comment. We revised these errors.

5. Table 4 "impaired cognitive function" is written twice in the legend.

Thank you for your comment. We corrected this error.

---

## [Editor Report · Decision Letter 2]

7 Sep 2020

Analysis of clinical outcomes in elderly patients with impaired swallowing function

PONE-D-20-09840R2

Dear Dr. terai,

We’re pleased to inform you that your manuscript has been judged scientifically suitable for publication and will be formally accepted for publication once it meets all outstanding technical requirements.

Kind regards,

Jason Chia-Hsun Hsieh, M.D. Ph.D

Academic Editor

PLOS ONE

Additional Editor Comments (optional):

All the questions were answered adequately.
---

## [Editor Report · Acceptance letter]

10 Sep 2020

PONE-D-20-09840R2 

Analysis of clinical outcomes in elderly patients with impaired swallowing function 

Dear Dr. terai:

I'm pleased to inform you that your manuscript has been deemed suitable for publication in PLOS ONE. Congratulations! Your manuscript is now with our production department. 

Kind regards, 

on behalf of

Dr. Jason Chia-Hsun Hsieh 

Academic Editor

PLOS ONE